# Microwaves effectively examine the extent and type of coking over acid zeolite catalysts

B. Liu[1], D.R. Slocombe[1,2], J. Wang[3], A. Aldawsari[1], S. Gonzalez-Cortes[1], J. Arden[1], V.L. Kuznetsov[1], H. AlMegren[4], M. AlKinany[4], T. Xiao[1] & P.P. Edwards[1]

Coking leads to the deactivation of solid acid catalyst. This phenomenon is a ubiquitous problem in the modern petrochemical and energy transformation industries. Here, we show a method based on microwave cavity perturbation analysis for an effective examination of both the amount and the chemical composition of cokes formed over acid zeolite catalysts. The employed microwave cavity can rapidly and non-intrusively measure the catalytically coked zeolites with sample full body penetration. The overall coke amount is reflected by the obtained dielectric loss ($\varepsilon''$) value, where different coke compositions lead to dramatically different absorption efficiencies ($\varepsilon''$/cokes' wt%). The deeper-dehydrogenated coke compounds (e.g., polyaromatics) lead to an apparently higher $\varepsilon''$/wt% value thus can be effectively separated from lightly coked compounds. The measurement is based on the nature of coke formation during catalytic reactions, from saturated status (e.g., aliphatic) to graphitized status (e.g., polyaromatics), with more delocalized electrons obtained for enhanced Maxwell–Wagner polarization.

[1] KACST – Oxford Petrochemical Research Centre (KOPRC), Inorganic Chemistry Laboratory, Department of Chemistry, University of Oxford, Oxford OX1 3QR, UK. [2] School of Engineering, Cardiff University, Queen's Buildings, The Parade, Cardiff CF24 3AA, UK. [3] Department of Materials, University of Oxford, Oxford OX1 3PH, UK. [4] Petrochemical Research Institute, King Abdulaziz City for Science and Technology, P.O. Box 6086, Riyadh 11442, Saudi Arabia. Correspondence and requests for materials should be addressed to T.X. (email: xiao.tiancun@chem.ox.ac.uk) or to P.P.E. (email: peter.edwards@chem.ox.ac.uk)

Zeolite catalysts, such as acid zeolite H-Y, H-ZSM-5, and H-SAPO-34 are widely employed across the modern petrochemical and fine chemical industries, for their excellent catalytic performance in hydrocarbon conversions and related chemistries[1, 2]. However, progressive product/intermediate accumulation in/on these porous catalysts followed by inevitable dehydrogenation into the coke deposits always leads to catalyst deactivation, necessitating catalyst replacement to maintain the reaction activity and minimize production interruptions[3–6].

For zeolite catalyzed hydrocarbon conversions, conjugated olefins, polyaromatics, and other pre-graphite (deeply dehydrogenated, $sp^2$ carbon rich) species are the major components in coke deposits and therefore allow the detection of their existence via electromagnetic (EM) spectroscopy on targeted chemical structures, e.g., C = C bond in an olefin molecule[4, 6–10]. Fourier-transform infrared (FT-IR) spectroscopy[11], better applied for in situ studies[12], plus Laser-Raman (optimized by Vis to UV)[13, 14] have been employed to draw a complementary picture of the various coke compounds. However, problems caused by complex sample preparation (e.g., when IR samples need to be mixed with KBr), atmospheric contamination of samples (e.g., moisture in air) and rigid temperature requirements greatly limit the widespread application of these measurements[8, 10]. More importantly, one common feature of the above spectroscopies, is a beam to sample working mode applied to coke analysis, according to which the EM wave propagates to a single point on the sample surface then transmission or reflectance data are recorded. Disadvantages include a localized, single point interrogation and the lack of information from internal coke species (deposited inside the zeolite cavity/channel) due to the limitation of energy penetration (here incident radiation needs to pass through the material layers to reach the inner part of the sample, in which process energy is gradually absorbed, and a thin film of sample is normally required)[4, 8, 10, 11, 13, 14]. On the contrary, nuclear magnetic resonance (NMR) spectroscopy works in a sample inside EM field mode, where the sample is instead bathed in the EM (i.e., magnetic) field for measurement with the entire sample body volumetrically interrogated (not focused on a single point on the sample surface). Theoretically, for such mode field penetration to each part of the sample is allowed, where sample thickness is no longer the major technical issue, thus, this is better for analysis of cokes located deeply inside the zeolite structures. However, problems of $^{13}C$ NMR in coke analysis arise from the higher cost and low efficiency of $^{13}C$ isotope exchange; and in the case of cross polarization (CP) without $^{13}C$ exchange, the signal of deeply dehydrogenated coke species is extremely poor due to the dissipation of hydrogen in substance[15, 16]. To overcome the above shortcomings we resort to using the microwave cavity perturbation technique which also enables sample interrogation in an EM field, and can show the growth of carbonaceous species quantitatively, as well as the dielectric property change of the whole catalyst body even in situ[17, 18].

Here, we show a microwave cavity perturbation based method to effectively measure the coke accumulation in the whole structure of an acid zeolite catalyst (volumetrically), and separate different coke compositions. We have shown that different coking levels (they have different coke accumulations) of acid zeolite catalysts can be readily distinguished by their dielectric loss properties, as reflected in their different $\varepsilon''$ values probed by the microwave cavity perturbation technique. The contribution to integral dielectric loss value of a coked sample by unit weight of cokes, given by $\varepsilon''$/wt%, is entirely characteristic of the coke composition formed under different reaction conditions. Particularly, we find that at the working frequencies near 2.45 GHz, polyaromatics dominate in the microwave response, with outstanding $\varepsilon''$/wt% values, as compared to olefin/paraffin cokes. The

observed results correspond closely with data obtained from previous coke characterization methods, e.g., Raman, thermogravimetric analysis (TGA) and $^{13}C$ NMR. The present technique possesses distinct advantages in terms of volumetric measurement with sample full body penetration, and higher sensitivity for deeply dehydrogenated cokes. This advance could provide critical information for monitoring catalyst coking and deactivation in important industrial processes (e.g., an industrial fluid catalytic cracking (FCC) process for petroleum refinery). The microwave-based approach interrogates the nature of catalytic coke formation which is an evolution from $sp^3$ carbons to $sp^2$ carbons that possess a further delocalized bond electron distribution, i.e., from saturated alkanes/olefins to the coke graphite structures with a conjugated $\pi$ electron system. By far, the available spectrum for catalyst analysis ranges from X-ray, to UV, Vis, and IR, and here our findings embody the potential to extend this to the microwaves.

## Results

**Our method measures samples with full body penetration.** Here a zeolite sample (un-reacted or coked) is placed into a quasi-static electric field (E-field) generated inside a microwave resonant cavity for measurement. The microwave electric field penetrates (interacts with) all of the volume and so the entire sample is interrogated. The results reveal the dielectric properties of the whole sample[18–20]. Such a whole-body, volumetric sample interrogation presents two advantages when applied to coke analysis. One important feature is that the measured data reflect the integrated coke accumulation of the entire sample, information are not obtained from a beam-focused point. The other is that the response derives from the fact that the effects (e.g., polarization) of the employed E-field work on each part of the sample body, and this fully penetrating feature can measure all coke contents even those deep in the zeolite porous structure[19]. More importantly, the technique is better suited to those deeper dehydrogenated coke species (giving an excellent signal response), thus it well covers the shortcomings of CP $^{13}C$ NMR.

The employed microwave cavity (Fig. 1) uses a $TM_{010}$ mode operating around 2.45 GHz[18, 19]. Schematic of representative resonant traces showing $|S_{21}|^2$ (transmitted microwave power) as a function of frequency is shown in Fig. 1a[19]. Sample insertion into the E-field antinode (this is exactly where the measurement is carried out) of the $TM_{010}$ mode causes a negative shift in the resonant frequency ($f$), and an increase in the bandwidth (BW). From the changes in resonant frequency ($\Delta f$) and bandwidth ($\Delta$BW) we can infer the dielectric properties (complex permittivity) of sample (all frequency and bandwidth measurements are adjusted to account for the presence of the quartz sample tube). Here, the complex permittivity ($\varepsilon^\star$) is defined as $\varepsilon^\star = \varepsilon' - j\varepsilon''$, where the real part ($\varepsilon'$) describes the polarization of material in response to the EM field and the imaginary part ($\varepsilon''$) describes the EM energy absorption (dielectric loss) in the sample[21, 22]. In this work, we focus on the imaginary permittivity, $\varepsilon''$, obtained with Eq. (1) adjusted from previous studies[20].

$$2\varepsilon'' A V_s = \frac{\Delta \text{BW}}{f_0} \qquad (1)$$

Here $f_0$ is the unperturbed resonant frequency. $A$ is a constant determined by the size and geometry of the cavity. For the present system, $A$ is $\sim 7.34 \times 10^{-3}$, as detected using a polytetrafluoroethylene (PTFE) sample of known complex permittivity[23]. $V_s$ is the effective volume of sample in the cavity (i.e., $\sim 0.126\ cm^3$)[19]. This is a non-destructive, non-invasive, and contact-less measurement, plus data acquisition takes only milliseconds and shows

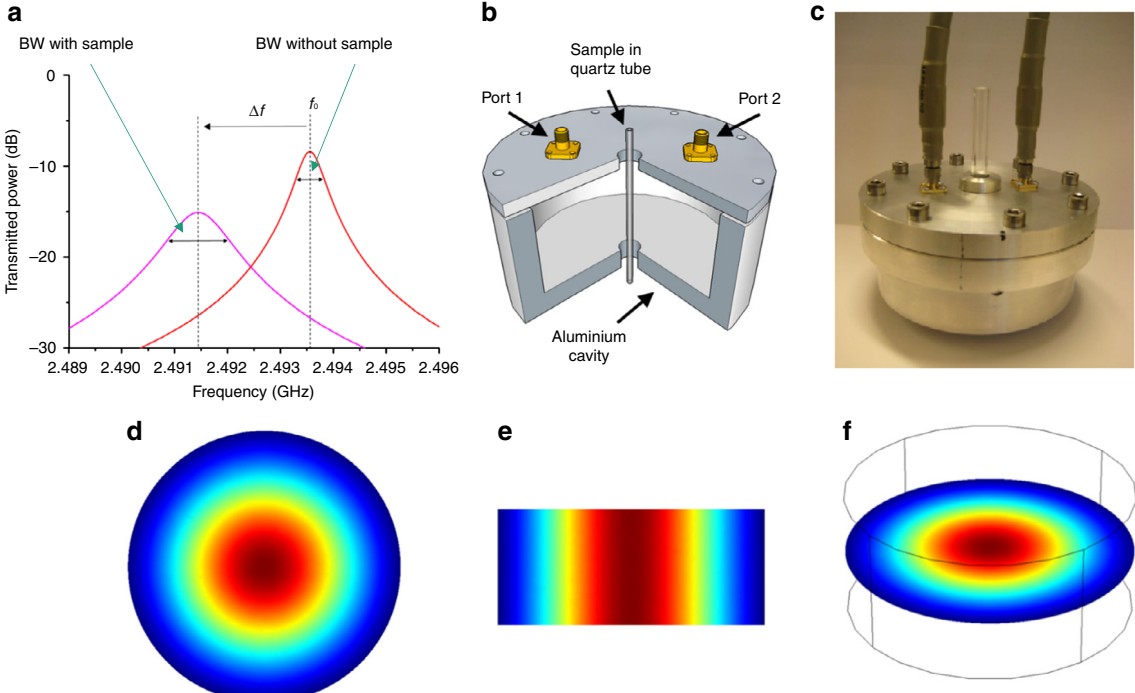

**Fig. 1** Sets and electric field distribution of the employed microwave cavity. **a** Schematic of example resonant traces showing the sample induced changes of microwave resonant cavity, which presents a shift in frequency and the change of bandwidth. **b** Quarter section of the perturbation cavity, cut to show the location of the sample tube (inner diameter 2 mm). **c** Exterior look of the microwave cavity. **d** Plan, **e** side and **f** slant views of the electric field distribution in the employed microwave resonant cavity

excellent repeatability among multiple tests. Besides, previous research has shown that these measurements can be taken at higher temperatures which would benefit future in situ applications and better contribute to the real-time monitoring of catalyst deactivation[17, 24].

The microwave cavity is designed in a cylindrical shape, as shown schematically in Fig. 1b. The sample was placed in a thin-walled high-purity quartz tube and introduced axially through a small insertion hole in the center of the top and bottom plates of the cavity (Fig. 1c). The cavity is made from aluminum with an unloaded quality factor ($Q$ factor) of ~8000 at room temperature. The internal dimensions of the cavity are radius $a = 4.6$ cm and length $d = 4.0$ cm. Here we choose $TM_{010}$ mode for the complex permittivity measurement of sample since it has a highly uniform E-field (the E-field antinode of this mode) near the cavity axis, resulting in minimal depolarization of sample in this experimental configuration[25]. The E-field is directed and parallel to the cavity axis, thus, parallel to the axis of the sample tube giving insignificant modification of the local electric field in the presence of the quartz tube. The distribution of the electric field magnitude is shown in Fig. 1d–f. For tubes of inner radii $r \ll a$, we may assume the electric field of the $TM_{010}$ mode to be in the small perturbation limit[19]. In our work, $r/a = ~0.022$, so the E-field remains highly uniform when applied to the sample. The radius $a$ is chosen so that the $TM_{010}$ mode has a resonant frequency of ~2.45 GHz (the exact working frequency may change within a small range in different tests and only causes a negligible difference). The cavity aspect ratio $d/a$ is tuned to be small enough for the $TM_{010}$ mode to be the dominant mode and clear of other, higher order modes in the cavity, but not too small to significantly compromise the high $Q$ factor or the axial uniformity of the electric field. Microwave measurements are carried out using an Agilent E5071B network analyzer. Using the $S_{21}$ scattering parameter, measurements of the transmitted

microwave power $|S_{21}|^2$ are taken and non-linear, least-squares curve fitting to a Lorentzian curve is used to extract the resonant frequencies and the loaded quality factors ($Q_L$). Microwaves are delivered to the cavity via a pair of SMA jack connectors, positioned in the top surface of the cavity, 3.0 cm from the axis. The open circuit terminations have an extended center conductor, which couples capacitively to the $TM_{010}$ mode electric field. We remove the effects of cavity couplings by converting $Q_L$ into the unloaded quality factor $Q$ in each case. The sample tubes were filled to a depth of 5 cm with the powder samples (4.0 cm of which will be in the active region of the cavity). The active sample volume is therefore 0.126 cm$^3$.

**Cavity response clearly reflects the coke accumulation.** Before all analyses, precise loading of the sample (~0.15 g) was required for minimal influence caused by the difference in sample weight. We initially tested a variety of zeolite types (QD Lianxin, China). Samples were coked in carefully controlled methanol-to-hydrocarbons (MTH) reactions where different reaction periods (2 h and 5 h, respectively) resulted in evolving different coke depositions for a given zeolite. Further investigations were carried out into three nano H-ZSM-5 zeolites (Si/Al ratios = 46, 60, and 160, ZEOLYST, USA) with various numbers of Brønsted acid sites. These zeolites with superior coke tolerance allow us to separate their post-run samples coked at the top part (top) and bottom part (bottom) of the catalyst bed (the reactor is set vertically with gas flow passing through from top to bottom), all achieved in the same 5 h MTH reaction. The resultant coke depositions along catalyst bed are illustrated in Fig. 2.

We have selected nano H-ZSM-5 (Si/Al = 160, marked as ZEO 160) samples for demonstration, which exhibit the most obvious visual color difference. The coked top sample, as shown in Fig. 2, is in darker black color (more coke deposits), whereas the coked

bottom sample only exhibits light gray color (less coke deposits), all in comparison with the fresh-white (no coke deposits) unreacted zeolite. Further, easily distinguished in transmission electron microscopy (TEM), the coked top sample possesses significantly larger black coking zones on the crystal surfaces (Fig. 2a), while those black areas are only randomly distributed on the zeolite surfaces of the coked bottom sample (Fig. 2b), all observed in multiple scans (corresponding TGA results are shown

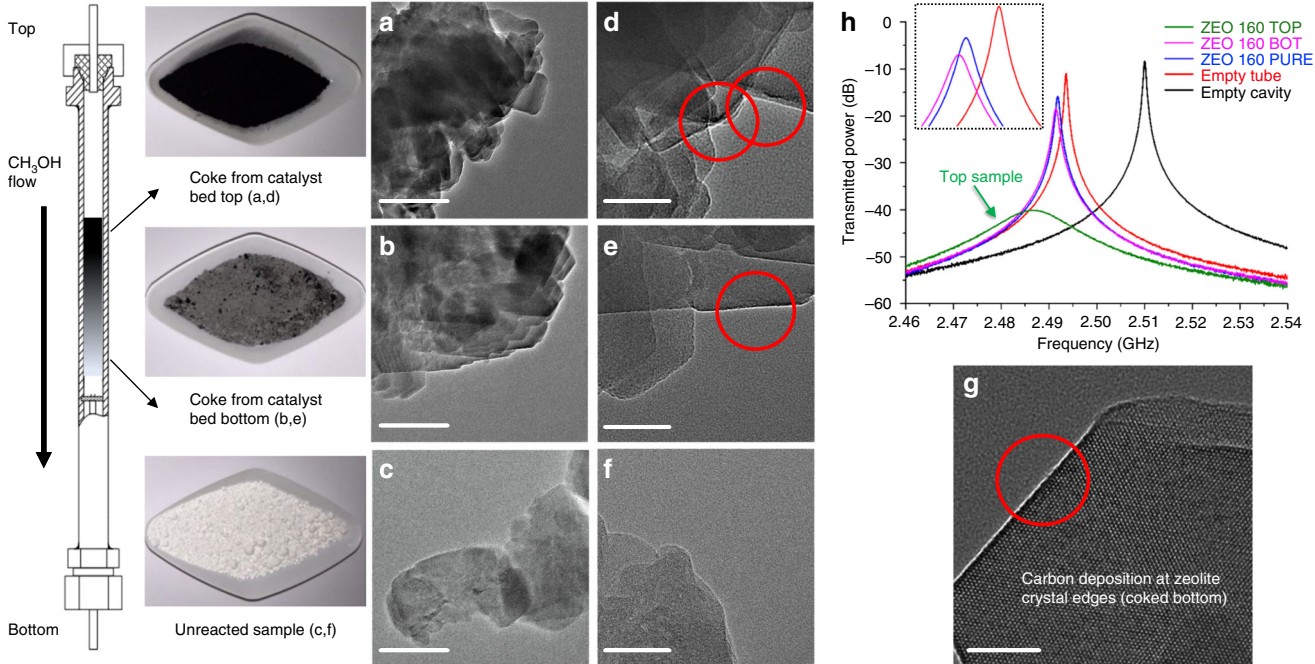

**Fig. 2** Coke depositions of ZEO 160 sample along the catalyst bed shown with their TEM pictures and microwave plots. **a–c** 200 nm vision transmission electron microscopy (TEM) images of coked top, coked bottom, and un-reacted samples. Overall 50 nm vision TEM images of coked top (**d**), coked bottom (**e**) and un-reacted (**f**) samples. Bulky powders and a schematic representation of the coke depositions in color gradient along the methanol flow route in the employed fixed bed reactor are shown on the *left*. **g** Amplified 20 nm TEM image implies carbon accumulation at the zeolite crystal edges. **h** Captured microwave plots induced by different samples (ZEO 160 group) and conditions. Catalyst unloading started from the reactor top by removing the top nut carefully. *Black*, deeply-coked samples (*top*) were collected firstly, until the lightly coked, *gray color samples* (*bottom*) emerged. The word BOT in **h** is an abbreviation for bottom. *Scale bar*: **a–c** 200 nm; **d–f** 50 nm and **g** 20 nm

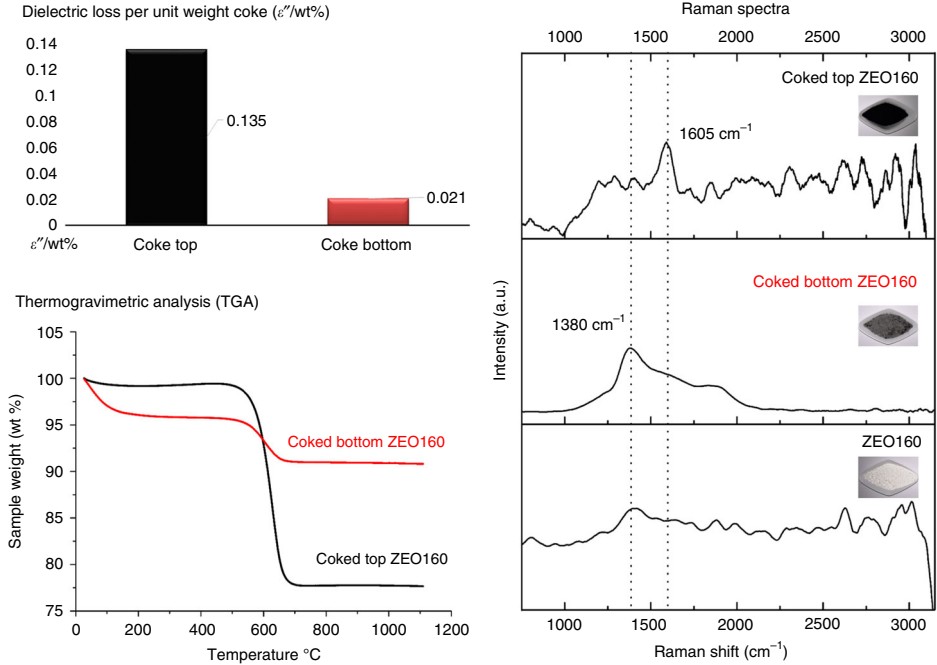

**Fig. 3** ε″/wt% values of differently coked ZEO 160 samples with supporting TGA and Raman results. ε″/wt% was calculated for coked top ZEO 160 and coked bottom ZEO 160 samples, respectively. TGA (physic & chemical approaches) indicates a total coke amount (wt% of post-run sample) without specification, whereas Raman (spectroscopy methods) helps in coke species identification but lacks of quantification information

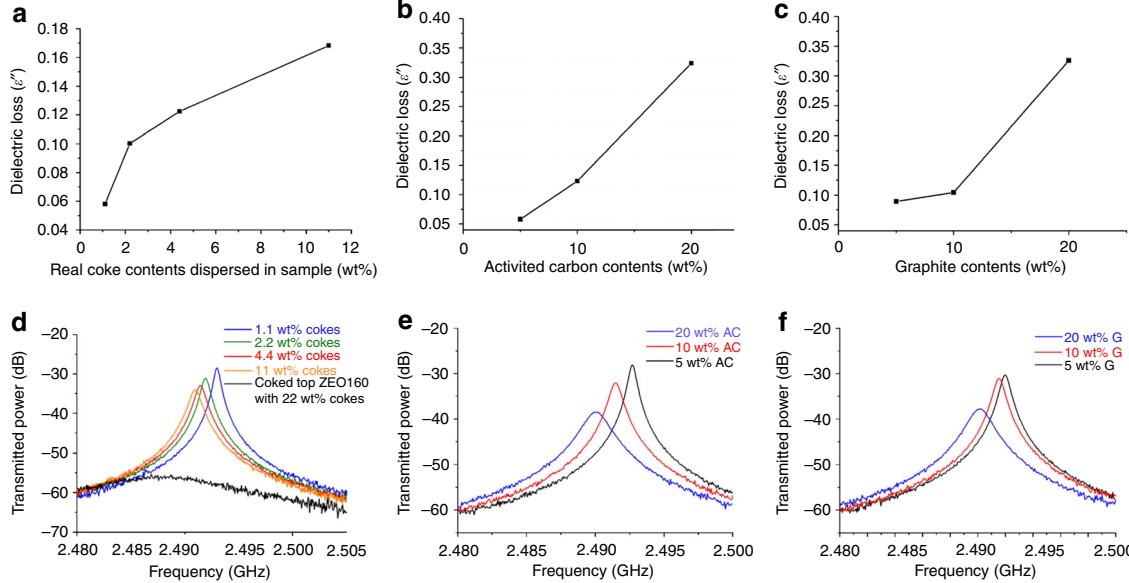

**Fig. 4** Dielectric loss values of different carbon-mixed samples as a function of increasing the amount of carbons. Dielectric loss values: **a** Coked top ZEO 160 sample mixed with fresh pure ZEO 160. **b** Activated carbon mixed with fresh pure ZEO 160. **c** Graphite mixed with fresh pure ZEO 160. Corresponding microwave plots: **d** Coked top ZEO 160 sample mixed with fresh pure ZEO 160. **e** Activated carbon mixed with fresh pure ZEO 160. **f** Graphite mixed with fresh pure ZEO 160

in Fig. 3). Darker areas are also observed on pure zeolite (Fig. 2c), probably due to the overlapping of crystals. However, under the vision of higher magnification only coked zeolites (Fig. 2d, e) show some regularly continuous clear signs of superficial carbon deposition at the non-overlapping edges of crystals (Fig. 2g), which is consistent with the results reported previously[26]. Again, the coked top sample (Fig. 2d) shows more intensively blacking edge regions (this again indicates more cokes are formed over this sample). One notes that this increase in coking resulted in the top sample easily distinguished from the others in the resulted microwave plots, as shown by the greatly broadened and shifted resonant bands with greater associated insertion loss (light green) in Fig. 2h. For calibration, the empty quartz tube also leads to a small shift of the resonant frequency (red), which is accounted for in the measurements of $\Delta BW$ and $\Delta f$ though dielectric losses associated with the sample tube are low, thus its $\Delta BW$ is nearly zero. The pure zeolite sample (blue) shows small $\Delta f$ and $\Delta BW$, and this is also the case (purple) for mildly coked bottom sample (see the magnified inset in Fig. 2h). Note that the apparently poor resolution of the GHz scale in Fig. 2h does not reflect the actually very high resolution of the measurement itself ($\sim 10^{-6}$ GHz).

**Absorption efficiency specifies the coke composition.** Exact dielectric loss values ($\varepsilon''$) of coked samples were calculated from the recorded $\Delta BW$ and $f_0$, as shown in Supplementarys Figs. 1 and 2 (Supplementary Information are not presented in the main text and attached as supporting materials). Corresponding Raman and TGA data were used as references to evaluate the coke compositions and quantities (Supplementary Figs. 3–11). These data illustrate that microwaves can quickly and accurately measure the increase in coke contents. Since the cavity perturbation measurements are volumetric, we can normalize the measured $\varepsilon''$ values by the weights of cokes formed over corresponding samples, as obtained from the TGA results (weights of different samples measured in tube are quite close and only cause negligible difference, so wt% from TGA is directly employed instead of the real coke weight in grams, also for a convenience of TGA), so as to build up standard calibration profiles. The calculated data ($\varepsilon$

$''$/wt%) reflects the contribution of unit weight cokes to the obtained, integral sample dielectric loss value $\varepsilon''$ (Supplementary Note 1), which is characteristic of the sample's coke composition. It provides a unique identifier for different types of coke deposited, since $sp^3$ and $sp^2$ type carbons will have dramatically different values.

Again, samples of ZEO 160 are chosen for demonstration with all information ($\varepsilon''$/wt%, Raman and TGA) included in Fig. 3. Cokes in the upper (top) part of the catalyst bed are mainly polyaromatics (a characteristic $\sim$1605 cm$^{-1}$ band in Raman spectra) in a large amount, while the post-run bottom sample shows a much lower coke deposition, with mildly enhanced Raman signals at bands between 1300 and 1550 cm$^{-1}$, partially overlapped with the zeolite framework and most possibly assigned to some olefinic, or other aliphatic deposits (less dehydrogenated) as precursors to the heavier coke species[13, 27]. As shown in Fig. 3, an apparently higher $\varepsilon''$/wt% value (0.135) is achieved by those deposited polyaromatic species which are predominant only in the top ZEO160 sample but hardly observable in the bottom ZEO160 sample. Similar results have been observed in other sample groups (Supplementary Figs. 12 and 13). These samples all experienced the same MTH conditions, but vary in coke composition, due to different zeolite types, reaction time periods, or loading positions in the reactor. Here our investigations reveal that polyaromatics lead to a higher dielectric loss value when present in large portions in a sample. Olefins, paraffins and other less-dehydrogenated species contribute to only limited dielectric loss values and when present in large proportions in coke contents can be readily separated due to their much lower $\varepsilon''$/wt% values (e.g., 0.021 in Fig. 3, see also Supplementary Note 2). Therefore, by comparing the calculated $\varepsilon''$/wt% values, different coke compositions can be separated. In a real reaction process, the coke compounds continuously evolve (e.g., from olefins to aromatics) as they are accumulated, therefore the difference in $\varepsilon''$ values, as a product of dielectric loss efficiency ($\varepsilon''$/wt%) and the weight (wt% in our method) of cokes, is much more apparent (e.g., top ZEO160 vs. bottom ZEO160 gives 2.979:0.134). Notably, the deeper dehydrogenated (graphitized) status of polyaromatics is a key characteristic for their apparently

higher dielectric loss, due to a rich $sp^2$ carbon abundance[8]. This is borne out by the very poor or even invisible CP $^{13}$C NMR responses on those samples (polyaromatics rich) with higher $\varepsilon''$/wt% values, which reflect the high deficiency of hydrogen atoms (Supplementary Figs. 14–27). Hydrocarbon coking loses hydrogen and forms more aromatic rings made up of $sp^2$ carbons (this is the typical structure of ultimate coke compounds, i.e., graphite)[8]. These aromatic $sp^2$ carbons have more conjugated and further delocalized bond electron distribution than the $sp^3$ carbons and $sp^2$ carbons in non-aromatics (e.g., olefins), which possesses highly mobile $\pi$ electrons that are able to undergo Maxwell–Wagner polarization to a greater extent (the polarization brings about separation of charges which generates local currents in substance and the resultant electron scattering processes cause dielectric loss in terms of heat), and therefore leads to apparently enhanced dielectric loss performance (reflected by $\varepsilon''$/wt%, see Supplementary Note 3)[19].

**The method is verified using carbons–zeolite mixtures**. For verification of the above, we further measured three sample groups carefully prepared by mechanically mixing different carbon sources with the fresh pure ZEO160 sample. The carbon sources include the coked top ZEO160 sample (5, 10, 20, and 50 wt% of coked top ZEO160 sample diluted in pure ZEO160, equal to 1.1, 2.2, 4.4, and 11 wt% real cokes in the mixture, as TGA shows coked top ZEO160 contains ~22 wt% cokes), activated carbon (5, 10, and 20 wt%), and graphite (5, 10, and 20 wt%). Measured results (Fig. 4) confirm that solid carbon (activated carbon and graphite) dispersions in zeolite lead to large increase of the sample dielectric loss value $\varepsilon''$ as detected by our method. Besides, the $\varepsilon''$ value also arises as a function of expanding the same coke constitution (coked top ZEO160 sample increases in portions, wt%, in the mixtures). All the samples are well separated in the captured plots. Notably, for mixtures with carbons, only at 5 wt% and above a well-developed linear increasing trend of $\varepsilon''$ value can be achieved. In contrast, the linear increase of real coke contents in zeolite can be detected at a much lower level (starting from 1.1 wt%). We note that the undiluted coked top ZEO160 sample with 22 wt% real coke contents exhibits a much higher $\varepsilon''$ value ( ~ 2.979), not in a same magnitude, than all the mechanically diluted samples. This indicates that real coke deposits formed and dispersed naturally in the zeolite structure during reactions possess much higher microwave absorption efficiency than those mechanical mixtures (even mixing the same coked sample with pure zeolite cannot achieve the same effect). The most possible reason could be a more uniform, contiguous, thinner-layer dispersion of carbon in the zeolite system[28], and can only be achieved by the conditions of a catalytic reaction, or the interactions between the coke species and zeolite frameworks.

## Methods

**General**. Microwave equipment settings are discussed in the main text.

The catalyst tests were carried out with a fixed-bed-reactor system. Each time, 1.0 g of sample was loaded in the middle of the tubular reactor, supported by carborundum particles (Fisher, 24 grit). Methanol (Sigma, reagent standard) was injected by a HPLC pump, and preheated to 150 °C to fully vaporize into gas phase. An enhanced Weight-Hourly-Space-Velocity of 8 h$^{-1}$ was applied (i.e., every 1 h, 8 g of methanol were passed over 1 g of catalyst) in the reaction. This heavy methanol conversion duty than usual was designed for a faster coke formation, and resulted in distinguished coke compositions over different samples, owing to the variety in zeo-type, reaction length, as well as sample position in the catalyst bed. N$_2$ was used as carrier gas to bring the vaporized methanol into reactor (5 ml min$^{-1}$). The catalyst bed temperature was set at 450 °C under atmosphere pressure, for the maximum coking performance within the 5 h period. The catalyst testing system is shown in Supplementary Fig. 28.

The post-run samples were carefully unloaded from the tubular reactor, after cooled down to the room temperature. For the nano zeolites, the firstly pour-out 0.4 g catalyst powders were collected precisely and marked as the coked catalyst-bed-top sample, whereas the later pour-out 0.5–0.6 g catalyst powders were saved

as the coked catalyst-bed-bottom sample. Grindings for 3 min were employed before the characterizations on each sample for the best mixing effects and a fair result.

Physical mixtures of carbons and zeolite were obtained by adding the solid samples (coked top ZEO 160/activated carbon/graphite) into fresh pure ZEO160 zeolite in pre-calculated amounts, respectively, then grinding for 3 min until the color of mixture became uniform.

Raman spectra were recorded on a Perkin-Elmer Raman Station 400F Raman Spectrometer. The samples were supported on a piece of clean glass for scanning.

A SDT Q600 (TA instruments) thermogravimetric analyzer was used to assess the coke content over the used catalysts. The coke amount was measured by the weight loss of coked samples during temperature-programmed calcination in air from 20 to 1120 °C (temp. ramp 10 °C min$^{-1}$). An aliquot of 50 mg of spent sample was used each time.

$^{13}$C NMR experiments were carried out on a Varian VNMRS spectrometer at ambient temperature with a resonance frequency of 100.562 MHz. CP technique was employed. The $^{13}$C CP NMR used a 6 mm probe, with data acquisition in 30.0 ms, recycle delay of 2.0 s and sample spinning rate at 6 KHz.

TEM measurements were undertaken using JEM-3000F microscope (300 kV). Samples were dispersed in ethanol and baked out in vacuum after transferring onto 300-mesh copper TEM holey carbon grids.

Some more discussions on the Supporting materials as Supplementary Information are included in Supplementary Discussion. Supplementary Table 1 is employed to explain the abbreviations in Supplementary Figures.

**Data availability**. The authors declare that the data supporting the findings of this study are available from the authors upon reasonable request.

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

## Acknowledgements

We would like to thank our colleagues of the PPE group at Oxford, and financial support from KACST, Saudi Arabia. We are also grateful to the careful characterization using NMR by Dr. David Apperley et al. at Durham University (of the EPSRC). Dr. Apperley also helped to explain why an aromatic rich sample would have a poorer CP $^{13}$C NMR response when there is no $^{13}$C exchange. Also, we thank Prof. Roger Y. Tsien (University of California, San Diego, USA) and Sir John Meurig Thomas (University of Cambridge, UK) for helpful discussions during their visits to Oxford and subsequent communications.

## Author contributions

B.L. conceived the study, designed the MTH reactions and also prepared the coked zeolite samples. He later performed the MW measurements as well as other characterizations, and finished the data treatments and analyses, with which he completed most of the manuscript. D.R.S. designed the MW cavity and was the leader in system set-up and adjustment, who was a senior researcher in MW absorption. P.P.E. (FRS) was the group leader at KOPRC and he supervised the above work. T.X. also supervised the project, with the initial idea sparked in his mind. Both the corresponding authors have put their great efforts into the revising work of the manuscript. J.W. is a third year PhD student in the Materials Department of Oxford, who performed all the TEM studies. A.A., S.G.-C., J.A., and V.L.K. are the key members of PPE group at Oxford who donated important jobs in the catalyst testings, characterizations as well as paper writings. H.A. and M.A. are the group leaders of KACST and they took part in the important works in the reaction design and their organization has funded the study.

## Additional information

**Competing interests:** The authors declare no competing financial interests.

