## [Peer Review File · Nature Communications]

Editorial Note: In their review of the first version of this manuscript, reviewer #2 added their comments to the manuscript file. These comments, excluding minor textual revisions, have been copied into this Peer Review File.

REVIEWERS' COMMENTS:

Reviewer #1 (Remarks to the Author):

This is a sound paper dealing with the analysis of coke formation occurring in zeolites. The dielectric response is clearly a measure for the degree of coke in the zeolite. The paper is well written and it should be published.

The idea to determine the coke in zeolites by microwaves is unique, however, when searching "(coke AND formation AND catalyst AND microwave)" in Scopus, I found N. Müller et al., "Initial tests to detect quantitatively the coke loading of reforming catalysts by a contactless microwave method", Chemical Engineering and Processing: Process Intensification, 50(8), pp. 729-731. It appears that this study and later papers of this (and probably other groups) use a similar method to measure the extent of coke formation; not on zeolites but on fixed-bed catalysts and automotive filters. Nevertheless, the paper provides enough novelty to justify publication.

English is used without obvious errors, and the article contains a common thread.

The results seem to be sound and reliable.

Reviewer #2 (Remarks to the Author):

Suggested changes:

Page 2: change "The microwave cavity perturbation also enables sample interrogation in an electromagnetic field; it can show the growth of carbonaceous species in numbers, and the dielectric property change of whole catalyst body in situ" to "*To overcome the above shortcomings we resort to using the microwave cavity perturbation technique which also enables sample interrogation in an electromagnetic field and can show the growth of carbonaceous species in numbers, and the dielectric property change of the whole catalyst body in situ*".

Page 2: "Here we applied this technique" to "*In this paper we have applied this technique*".

Remove italics style from " ϵ " throughout the manuscript.

Page 2: change V_s to V_s

Page 3: change "For the employed system, the A value is" to "*For the present system, A is*"

Page 3: change "that these measurements can be taken at higher temperatures, such benefits would help in the future in-situ applications" to "that these measurements can be taken at higher temperatures *which would benefits* in the future in-situ applications".

Page 3: change “The cavity is made from aluminium and we have obtained an unloaded quality factor” to “The cavity is made from aluminium and we have obtained *with* an unloaded quality factor”.

Page 3: change “The distribution of electric field magnitude” to “The distribution of *the* electric field magnitude”.

Page 3: change “may change in a small range in different tests” to “may change *within* a small range in different tests”.

Page 3: change “is used to determine resonant frequencies and loaded quality factors” to “is used to determine *the* resonant frequencies and *the* loaded quality factors”

Page 4: change “Before all analyses, precise loading of sample” to “Before all analyses, precise loading of *the* sample”.

Page 4: change “Also, easily distinguished in microscopy (TEM)” to “*Further*, easily distinguished in microscopy (TEM)”

Page 4: change “in the obtained microwave plots” to “in the resulted microwave plots”.

Page 6: split sentence “e. For verification of the above, we further measured 3 sample groups carefully prepared by mechanically mixing coked top ZEO160 sample (5wt%, 10wt%, 20wt% and 50wt% of coked sample diluted in fresh ZEO160, equal to 1.1wt%, 2.2wt%, 4.4wt%, and 11wt% real cokes, as TGA shows ~22wt% of coke contents in the undiluted coked sample), activated carbon (5wt%, 10wt% and 20wt%) and graphite (5wt%, 10wt% and 20wt%) with fresh pure ZEO160 zeolite, respectively (Fig. 4)” into two sentences.

Page 6: change “dispersions in zeolite lead to dramatic increase” to “dispersions in zeolite lead to large increase”

Page 6: split sentence “This indicates that real coke deposits formed and dispersed naturally in the zeolite structure during reactions possess much higher microwave absorption efficiency than those mechanical mixtures (even mixing the same coked sample with pure zeolite cannot achieve the same effect), which are most possibly attributed to a more uniform, contiguous, thinner-layer dispersion of carbon in the zeolite system²⁹, and can only be achieved by the conditions of a catalytic reaction, or the interactions between the coke species and zeolite frameworks (of course, this has sparked our interest for a further study in the future).” into two sentences.

Reviewers' comments:

Reviewer #1:

This is a sound paper dealing with the analysis of coke formation occurring in zeolites. The dielectric response is clearly a measure for the degree of coke in the zeolite. The paper is well written and it should be published.

The idea to determine the coke in zeolites by microwaves is unique, however, when searching “(coke AND formation AND catalyst AND microwave)” in Scopus, I found N. Müller et al., “Initial tests to detect quantitatively the coke loading of reforming catalysts by a contactless microwave method”, *Chemical Engineering and Processing: Process Intensification*, 50(8), pp. 729-731. It appears that this study and later papers of this (and probably other groups) use a similar method to measure the extent of coke formation; not on zeolites but on fixed-bed catalysts and automotive filters.

Nevertheless, the paper provides enough novelty to justify publication.

English is used without obvious errors, and the article contains a common thread.

The results seem to be sound and reliable.

Reply to Reviewer #1:

We sincerely appreciate the Reviewer 1 for these very encouraging comments. His/her understanding on our work is admirable and very professional, also the reviewer has highlighted that dielectric response is clearly a measure for the degree of cokes in zeolite structure which is the critical, central element of the study.

As the Reviewer 1 has noted, our novelty is based on our studies on zeolites, which is the most important and widespread catalysts in the petrochemical industry with intensive coke formation in any operating plant. Furthermore, we have developed the method to enable it to separate different coke compositions with our designed calibration profiles.

Reviewer #2:

PAGE 2

Question 1: The Reviewer 2 has added a sentence before introducing the microwave cavity perturbation technique in the Introduction of this paper. He/she also made some necessary changes on the words and grammars used in this paragraph. (Page 2, right, line 21)

These are reproduced below:

To overcome the above shortcomings we resort to using t The microwave cavity perturbation technique which also enables sample interrogation in an electromagnetic field and- it can show the growth of carbonaceous species in numbers, and the dielectric property change of the whole catalyst body *in situ*^{17,18}. Here- In this paper we have applied this technique to show coke accumulation on/in acid zeolite catalysts, and innovatively *separate different coke compositions*. This advance could provide critical information for monitoring catalyst coking and deactivation in important industrial processes (e.g. an industrial FCC process for petroleum refinery).

Reply: We gratefully accept the Reviewer 2's corrections and have made the necessary changes. The new contents have been shown below.

“To overcome the above shortcomings we resort to using the microwave cavity perturbation technique which also enables sample interrogation in an electromagnetic field, and can show the growth of carbonaceous species in numbers, as well as the dielectric property change of the whole catalyst body even in-situ^{17,18}.”

According to the *Nature Communications* template, we have to start the final paragraph in Introduction, by starting with “Here, we show...”, so we have combined the Reviewer 2's suggestions with the Editor's:

“Here, we show a microwave cavity perturbation based method to effectively measure the coke accumulation in the whole structure of an acid zeolite catalyst (volumetrically), and separate different coke compositions. We have shown that

different coking levels (they have different coke accumulations) of acid zeolite catalysts can be readily distinguished by their dielectric loss properties, as reflected in their different ϵ'' values probed by the microwave cavity perturbation technique. The contribution to integral dielectric loss value of a coked sample by unit weight of cokes, given by $\epsilon''/\text{wt}\%$, is entirely characteristic of the coke composition formed under different reaction conditions. Particularly, we find that at the working frequencies near 2.45GHz, polyaromatics dominate in the microwave response, with outstanding $\epsilon''/\text{wt}\%$ values, as compared to olefin/paraffin cokes. The observed results correspond closely with data obtained from previous coke characterization methods, e.g. Raman, TGA and ^{13}C NMR. The present technique possesses distinct advantages in terms of volumetric measurement with sample full body penetration, and higher sensitivity for deeply dehydrogenated cokes. **This advance could provide critical information for monitoring catalyst coking and deactivation in important industrial processes (e.g. an industrial FCC process for petroleum refinery).** The microwave based approach interrogates the nature of catalytic coke formation which is an evolution from sp^3 carbons to sp^2 carbons that possess a further delocalized bond electron distribution, i.e., from saturated alkanes/olefins to the coke graphite structures with a conjugated π electron system. By far, the available spectrum for catalyst analysis ranges from X-ray, to Ultraviolet, Visible, and Infrared, and here our findings embody the potential to extend this to the microwaves.”

PAGE 3

Question 2: The Reviewer 2 has suggested:

Changing all ϵ symbols into non-italics.

Removing the italics in $\epsilon^* = \epsilon' - j\epsilon''$.

Changing the Vs in equation $\epsilon''AVs = \frac{\Delta BW}{f_0}$ to V sub s.

These are reproduced below:

Here, the complex permittivity (ϵ^*) is defined as $\epsilon^* = \epsilon' - j\epsilon''$, edit to read $\epsilon' - j\epsilon''$ where the real part (ϵ') describes the polarization of material in response to the electromagnetic field and the imaginary part (ϵ'') describes the electromagnetic energy absorption (dielectric loss) in the sample^{21,22}. In this work, we focus on the imaginary

permittivity, ϵ'' , of the coked catalyst samples, obtained and adjusted from²⁰

$$2 \epsilon''AVs = \frac{\Delta BW}{f_0} \text{ change Vs to V sub s}$$

Reply: Many thanks! We have made the corresponding changes.

All ϵ symbols in the manuscript have been changed into non-italics.

The equations have been changed:

$$\epsilon^* = \epsilon' - j\epsilon''$$
$$2\epsilon''AV_s = \frac{\Delta BW}{f_0}$$

The Editor also made some suggestions on the formatting of symbols in this manuscript. “Please check proper formatting of symbols - see my e-mail for details. Scalar variables (e.g. x , V , χ) and constants (e.g. π , \hbar , e) should be typeset in italics, and vectors (such as r , the wavevector k , or the magnetic field vector B) should be typeset in bold without italics. In contrast, subscripts and superscripts should only be italicized if they too are variables or constants. Those that are labels (such as the 'c' in the critical temperature, T_c , the 'F' in the Fermi energy, E_F , or the 'crit' in the critical current, I_{crit}) should be typeset in roman”.

We have also made necessary changes by converting the following scalar variables and constants into the italics.

1) Scalar variables: S_{2l} (transmitted microwave power), f (frequency), f_0 (initial frequency of a sample), BW (bandwidth), Q (unloaded quality factor) and Q_L (loaded quality factor).

2) Scalar constants (they are not changed once the experimental configurations have been applied): V_s (sample volume), a (cavity radius), d (diameter), and r (tube inner radii).

The final manuscript formatting is a combination of suggestions from both the Reviewer and Editor.

Question 3: Some words need to be replaced on Page 3. (Page 3, right, line 6, line 18-19, line 30-31)

These have been replaced below:

Here f_0 is the unperturbed resonant frequency. A is a constant determined by the size and geometry of the cavity. For the ~~employed-present~~ system, ~~the A value~~ is approximately 7.34×10^{-3} , as detected using a PTFE sample of known complex permittivity²³. V_s is the effective volume of sample in the cavity (i.e. $\sim 0.126 \text{ cm}^3$)¹⁹. This is a non-destructive, non-invasive and contact-less measurement, plus data acquisition takes only milliseconds and shows excellent repeatability among multiple tests. Besides, previous researches have shown that these measurements can be taken at higher temperatures, ~~such which would benefit~~ ~~would help in the~~ future in-situ applications and better contribute to the real-time monitoring of catalyst deactivation^{17,24}.

The microwave cavity is designed in a cylindrical shape, as shown schematically in Fig. 1b. The sample was placed in a thin-walled high-purity quartz tube and introduced axially through a small insertion hole in the centre of the top and bottom plates of the cavity (Fig. 1c). The cavity is made from aluminium ~~and we have obtained with~~ an unloaded quality

Reply: Many thanks for the very careful corrections and a great patience! We have made the necessary changes and accepted the corrections.

“Here f_0 is the unperturbed resonant frequency. A is a constant determined by the size and geometry of the cavity. For the **present** system, A is approximately 7.34×10^{-3} , as detected using a PTFE sample of known complex permittivity²³. V_s is the effective volume of sample in the cavity (i.e. $\sim 0.126 \text{ cm}^3$)¹⁹. This is a non-destructive, non-invasive and contact-less measurement, plus data acquisition takes only milliseconds and shows excellent repeatability among multiple tests. Besides, previous research has shown that these measurements can be taken at higher temperatures **which would benefit future in-situ applications** and better contribute to the real-time monitoring of catalyst deactivation^{17,24}.”

“The microwave cavity is designed in a cylindrical shape, as shown schematically in Fig. 1b. The sample was placed in a thin-walled high-purity quartz tube and introduced axially through a small insertion hole in the centre of the top and bottom plates of the cavity (Fig. 1c). The cavity is made from aluminium **with** an unloaded quality factor (Q factor) of ~ 8000 at room temperature.”

PAGE 4

Question 4: Some grammar mistakes need to be corrected on Page 4, such as, adding “the” to the front of a noun.

Reply: Many thanks for the very careful corrections and a great patience! We have accepted the corrections. The revised texts have been shown below.

“The distribution of **the** electric field” (Page 4, left, line 10)

“the exact working frequency may change **within** a small range in ...” (Page 4, left, line 20)

“least-squares curve fitting to a Lorentzian response is used to determine **the** resonant frequencies and **the** loaded quality factors” (Page 4, left, line 36-37)

“Before all analyses, precise loading of **the** sample” (Page 4, right, line 13)

PAGE 5

Question 5: Some words need to be replaced or added on Page 5.

Reply: Many thanks for the very careful corrections and a great patience! We have made the necessary changes and accepted the corrections. The revised texts have been shown below.

“**Further**, easily distinguished in microscopy (TEM), the coked top sample...” (Page 5, left, line 4)

“in the **resulted** microwave” (Page 5, left, line 29)

“accounted for in **the** measurements of” (Page 5, right, line 1)

PAGE 6

Question 6:

On Page 6, the word “to” needs to be removed from the sentence “TGA is directly employed to instead of the real coke weight in grams...”. (Page 6, left, line 1)

“bot” for abbreviations is not suggested, instead, “bottom” should be used. (Page 6, right, line 31, the second line above the page bottom).

Reply: Many thanks for the very careful corrections and a great patience! We have accepted the corrections.

The corresponding corrections are shown below:

“TGA is directly employed instead of the real coke weight in grams...”

“top ZEO160 vs. **bottom** ZEO160”

Question 7: On Page 6, the Reviewer 2 noted that the Fig. S12-S13 were missing. (Page 6, right, line 5)

Reply: Many thanks! We have changed the references to Supplementary Figures by using the *Nature Communications* style. Now the two figures are referenced as “Supplementary Fig. 12 and 13”, they are included in the Supplementary Information. The new references will not confuse the readers with the figures in the main text and Supplementary Information.

PAGE 7

Question 7: One Page 7, the Reviewer 2 noted that the Fig. S14-S27 were missing. (Page 7, left, line 10)

Reply: Many thanks! We have changed the references to Supplementary Figures by using the *Nature Communications* style. Now the figures are referenced as “Supplementary Fig. 14-27”, they are included in the Supplementary Information. The new references will not confuse the readers with the figures in the main text and Supplementary Information.

Question 8: On page 7, “delimited currents” is confusing, and the Reviewer 2 has marked it with “?”. (Page 7, left, line 22)

These have been replaced below:

‘graphite’)⁸. These aromatic sp^2 carbons have more conjugated and further delocalized bond electron distribution than the sp^3 carbons and sp^2 carbons in non-aromatics (e.g. olefins), which possesses highly mobile π electrons that are able to undergo Maxwell-Wagner polarization to a greater extent (this process generates delimited?? currents and the resultant electron scattering processes cause energy loss in terms of heat), and therefore leads to apparently enhanced dielectric loss (reflected by $\epsilon''/\text{wt}\%$)¹⁹ *<Note.3>.

Reply: Many thanks! We apologize for the misleading texts. We have made the necessary changes so as the description is no longer confusing.

“These aromatic sp^2 carbons have more conjugated and further delocalized bond electron distribution than the sp^3 carbons and sp^2 carbons in non-aromatics (e.g. olefins), which possesses highly mobile π electrons that are able to undergo Maxwell-Wagner polarization to a greater extent (the polarization brings about separation of charges which generates local currents in substance and the resultant electron scattering processes cause dielectric loss in terms of heat), and therefore leads to apparently enhanced dielectric loss performance (reflected by $\epsilon''/\text{wt}\%$)¹⁹ *<Supplementary Note 3>.”

Question 9: The Reviewer 2 has suggested to split two very long sentences.

Reply: Many thanks! We have made the necessary changes according to the reviewer, and the contents are therefore better presented and well organized.

Long Sentence 1

“For verification of the above, we further measured 3 sample groups carefully prepared by mechanically mixing coked top ZEO160 sample (5wt%, 10wt%, 20wt% and 50wt% of coked sample diluted in fresh ZEO160, equal to 1.1wt%, 2.2wt%, 4.4wt%, and 11wt% real cokes, as TGA shows ~22wt% of coke contents in the undiluted coked sample), activated carbon (5wt%, 10wt% and 20wt%) and graphite (5wt%, 10wt% and 20wt%) with fresh pure ZEO160 zeolite, respectively (Fig. 4).”

The above sentence has been split below.

“(Sentence 1) For verification of the above, we further measured 3 sample groups carefully prepared by mechanically mixing different carbon sources with the fresh pure ZEO160 sample. (Sentence 2) The carbon sources include the coked top ZEO160 sample (5wt%, 10wt%, 20wt% and 50wt% of coked top ZEO160 sample diluted in pure ZEO160, equal to 1.1wt%, 2.2wt%, 4.4wt%, and 11wt% real cokes in the mixture, as TGA shows coked top ZEO160 contains ~22wt% cokes), activated carbon (5wt%, 10wt% and 20wt%) and graphite (5wt%, 10wt% and 20wt%).”

Long Sentence 2

“This indicates that real coke deposits formed and dispersed naturally in the zeolite structure during reactions possess much higher microwave absorption efficiency than those mechanical mixtures (even mixing the same coked sample with pure zeolite cannot achieve the same effect), which are most possibly attributed to a more uniform, contiguous, thinner-layer dispersion of carbon in the zeolite system²⁹, and can only be achieved by the conditions of a catalytic reaction, or the interactions between the coke species and zeolite frameworks (of course, this has sparked our interest for a further study in the future).”

The above sentence has been split below.

“(Sentence 1) This indicates that real coke deposits formed and dispersed naturally in the zeolite structure during reactions possess much higher microwave absorption efficiency than those mechanical mixtures (even mixing the same coked sample with pure zeolite cannot achieve the same effect). (Sentence 2) The most possible reason

could be a more uniform, contiguous, thinner-layer dispersion of carbon in the zeolite system²⁹, and can only be achieved by the conditions of a catalytic reaction, or the interactions between the coke species and zeolite frameworks (of course, this has sparked our interest for a further study in the future).”

Question 10: On Page 7, replace “dramatic” with “large”. (Page 7, left, line 47)

Reply: Many thanks! We have made the necessary changes according to the reviewer.

“Measured results (Fig. 4) confirm that solid carbon (activated carbon and graphite) dispersions in zeolite lead to **large** increase of the sample dielectric loss value ϵ ” as detected by our method.”

Reply to Reviewer #2:

Reviewer 2 has shown a powerful and very strong background in this area, with precise and very constructive suggestions on the paper. His/her work has helped to improve the quality of our paper by rigorous requirements on the labels, spellings, grammars, stylings and the organizing of words. The equations were also carefully checked and corrected.

We sincerely thank the Reviewer 2 for his/her great patience on revising our paper. The suggestions on structuring the paper, are based on the known-how in the related area (from the perspective of a senior expert in microwaves), and we believe that this contribution is crucial for a better quality of our paper.